# Explainable Reinforcement Learning Through Goal-Based Explanations

## Abstract

Deep Reinforcement Learning agents achieve state-of-the-art performance in many tasks at the cost of making them black-boxes, hard to interpret and understand, making their use difficult in trusted applications, such as robotics or industrial applications. We introduce **goal-based interpretability**, where the agent produces goals which show the reason for its current actions (reach the current goal) and future goals indicate its desired future behavior without having to run the environment, a useful property in environments with no simulator. Additionally, in many environments, the goals can be visualised to make them easier to understand for non-experts. To have a goal-producing agent without requiring domain knowledge, we use 2-layer **hierarchical agents** where the top layer produces goals and the bottom layer attempts to reach those goals.

Most classical reinforcement learning algorithms cannot be used train goal-producing hierarchical agents. We introduce a new algorithm to train these more interpretable agents, called **HAC-General with Teacher**, an extension of the Hindsight Actor-Critic (HAC) algorithm (Levy et al., 2019) that adds 2 key improvements: (1) the goals now consist of a state $s$ to be reached and a reward $r$ to be collected, making it possible for the goal-producing policy to incentivize the goal-reaching policy to go through high-reward paths and (2) an expert teacher is leveraged to improve the training of the hierarchical agent, in a process similar but distinct to imitation learning and distillation. Contrarily to HAC, there is no requirement that environments need to provide the desired end state. Additionally, our experiments show that it has better performance and learns faster than HAC, and can solve environments that HAC fails to solve.

## 1 Introduction

Deep learning has had a huge impact on Reinforcement Learning, making it possible to solve certain problems for the first time, vastly improving performance in many old problems and often exceeding human performance in difficult tasks (Schrittwieser et al., 2019; Badia et al., 2020). These improvements come at a price though: deep agents are black-boxes which are difficult to understand and their decisions are hard to explain due to the complexity and non-obvious behavior of neural networks. In safety-critical applications, it is often fundamental to check that certain properties are respected or to understand what the behavior of the agent will be (García & Fernández, 2015; Bragg & Habli, 2018). Simply observing the behavior of the agent is often not enough, since it might take its actions for the wrong reasons or it might have surprising behavior when faced with an unexpected state. Ideally, the agent would explain its behavior, which would allow for auditing, accountability, and safety-checking (Puiutta & Veith, 2020), unlocking the use of Reinforcement Learning systems in critical areas such as robotics, semi-autonomous driving, or industrial applications.

We provide three contributions to make more interpretable deep agents. First, we develop a new type of explanation for the agent's behavior. Imagine the following scenario: a robotic agent has to traverse a difficult terrain until it reaches a specific building, where it collects a reward. The agent decomposes its task into a series of goals (for example, positions it has to reach) and tries to reach these goals successively until it reaches the reward zone. The agent is more interpretable since it explicitly produces the successive goals it is trying to accomplish: the current goal explains its short-term behavior (the joint movements are done to reach the current goal position) and the remaining

goals help us understand the agent's overall plan to solve the task and predict its future behavior. We call **goal-based explanation** or goal-based interpretability the use of plan composed by a series of goals.

Both model-based reinforcement learning (Moerland et al., 2020) and planning techniques (Fox et al., 2017) appear similar to goal-based explanations but there are important differences that make this technique novel. Goal-based explanations do not require learning a model of the environment (neither the reward function nor the transition function), thus being compatible with both model-free and model-based reinforcement learning. Planning can be a useful explainability technique, but it has a few limitations: it typically requires knowing the end goals, they often cannot be applied to complex Markov Decision Problems and they may have difficulty handling very large or continuous action spaces or state spaces. Our approach suffers from none of these limitations.

Second, we develop a method to make the agent produce the goals that add interpretability. To do so, the agent is structured as a 2-level hierarchy of policies, with a goal-picking policy that produces goals and a goal-reaching policy that attempts to reach them. Goals are *(state, minimum desired reward)* pairs, meaning the goal-reaching policy has to reach a specific state in at most $H$ steps and collect a minimum amount of reward along the way. To create a goal-based explanation, the goal-picking policy is queried repeatedly: given the agent's state $s$, we query for the current goal $g_1 = (s_1, r_1)$; we then assume the agent reaches the state $s_1$ and query for the next goal $g_2 = (s_2, r_2)$; this process for a fixed amount of steps per environment, though in future work more sophisticated algorithms to determine the adequate number of goals could be compared.

Our third contribution is developing HAC-General, a new algorithm specifically designed to train goal-producing hierarchical agents. This algorithm builds upon the Hindsight Actor-Critic (HAC) algorithm (Levy et al., 2019) and makes it more widely applicable by not requiring the environment to provide an explicit end-goal. Instead of trying to reach the end-goal as fast as possible and ignoring the environment's rewards, the HAC-General algorithm trains the agent to maximize the collected reward. Our extension tries to preserve the key property that makes the Hindsight Actor-Critic algorithm effective: having an effective strategy to deal with non-stationarity by giving the illusion that the policies in sub-levels are optimal. The HAC-General algorithm is also able to leverage a black-box expert to improve and speed up the training for the hierarchical agent.

## 2    BACKGROUND & RELATED WORK

### 2.1    EXPLAINABLE REINFORCEMENT LEARNING

The Reinforcement Learning community has recognized the need for interpretable and explainable agents, and researchers have developed several methods to add explainability and interpretability. Puiutta & Veith (2020) survey explainability techniques; we briefly describe some key methods.

To add interpretability, *saliency-map methods* determine the importance of each input feature for the policy when it generates its output. Perturbation-based methods (Greydanus et al., 2018) measure importance by perturbing different parts of the input and measuring the change in the policy's output. The larger the change in output, the more important the feature; the magnitude of the change quantifies the relative importance of features, making it possible to build the saliency map. In object-based saliency maps (Iyer et al., 2018), in addition to measuring the importance of raw features, they also measure the importance of the whole objects present in the image. The importance of each object is measured by masking it and measuring the change in the policy's output. Thus, a higher-level object saliency map is created which can be more easily interpreted by non-experts.

Another approach is to *distill* the policy of the black-box agent into a simpler, more interpretable model while trying to preserve the behavior and performance of the black-box policy. Coppens et al. (2019) distill the black-box policy into a soft decision tree, a type of decision tree where the leaves output a static distribution over the actions and the inner nodes select the sub-branch using a logistic model. A different approach is taken by Liu et al. (2019) which distill the model into linear model U-Trees, a type of decision tree in which leaf nodes use a linear model to produce their output (Q-values) instead of outputting a constant value. Both types of decision trees are more interpretable since they follow clear and simpler rules to go down the tree and to pick the output value.

## 2.2 Hierarchical Reinforcement Learning

In Hierarchical Reinforcement Learning (HRL), an agent is composed of a hierarchy of policies. The top layer decomposes the task into sub-tasks, the layer below decomposes sub-tasks into sub-sub-tasks, and so on until the lowest level receives a low-level task and attempts to solve it by interacting with the environment. Policies at higher layers learn to act at higher temporal and abstraction levels.

A subtask $\phi^i$ can be defined in multiple ways, for example as simpler linearly solvable Markov Decision Problems (Earle et al., 2018) or as a tuple $(P^i, C^i_{comp}, R_i)$ where the subtask $\phi^i$ is eligible to start any time the precondition $P_i$ is satisfied and it is completed once the current state is part of the completion set $C^i_{comp}$ upon which it receives a reward $r^t \sim R_i$ (Sohn et al., 2020).

Our approach is based upon **goal-oriented** hierarchical reinforcement learning, where completing a task means reaching a goal where that goal typically is a state $s$ which the agent must reach. Policies that receive a goal have only $H$ steps to reach it instead of an unlimited time budget. The policy at the bottom of the hierarchy interacts with the environment while the other policies act by picking goals (i.e. their actions are goals for the policy below them). In some problem settings, the reward must be maximized. However, in other settings, the agent receives a goal $g_{env}$ from the environment which must be reached as fast as possible. In that setting, it is important to note that the agent ignores the rewards produced by the environment; it only uses its internal reward scheme which gives the agent a small negative reward at each step, encouraging it to find short paths.

While goal-oriented hierarchical reinforcement learning has a long history (Dayan & Hinton, 1992), there has been a resurgence in interest in recent years. Hierarchical-DQN (Kulkarni et al., 2016) combines hierarchical learning with deep learning for the first time; Hierarchical Actor-Critic (Levy et al., 2017) improves performance by carefully setting up the hierarchy of actor-critic policies; Deep Feudal Reinforcement Learning (Vezhnevets et al., 2017) use abstract goals in a latent space instead of an actual state in the real state space $\mathcal{S}$. More recently, Hierarchical Learning with Off-Policy Correction (Nachum et al., 2018) tries to support off-policy learning even though that all layers are constantly evolving by correcting the goals present in the transitions using a heuristic method.

While goals produced by feudal networks (Vezhnevets et al., 2017) might be more effective for training, they do not fit our interpretability objectives either since the goal space is a latent space that is not directly understandable to researchers and non-experts.

## 3 Generalized Hindsight Actor-Critic With Teacher

Our work builds upon the Hindsight Actor-Critic (Levy et al., 2019) or HAC, a state-of-the-art algorithm to train hierarchical agents, which achieves excellent performance in some environments beating the hierarchical agent algorithms mentioned before. We refer the interested reader to the original description of HAC (Levy et al., 2019) due to its non-trivial nature.

HAC is designed for a specific setting: environments that provide an end goal and where the only objective is to reach the goal as fast as possible. This specialization leads to **2 limitations**: (1) HAC requires a goal, making it incompatible with all environments which do not provide a goal for the agent and (2) HAC ignores the rewards given by the environment since it uses an internal reward scheme. This makes it inapplicable to most environments, in which rewards can be given anytime.

To address these issues we generalize HAC, creating the **HAC-General with Teacher** algorithm which doesn't require a goal and that considers the reward given by the environment. To avoid requiring a goal, the policy at the top of the hierarchy produces its output (a shorter-term goal) using only the state as input (no end-goal in the input). To take into account the rewards, the objective of the goal-picking policy becomes picking goals such that the maximum amount of reward is collected during the episode. The objective of policy at the bottom of the hierarchy (the goal-reaching policy) stays the same: reaching the short-term goal in at most $H$ steps, ignoring environment rewards.

### 3.1 Maintaining the optimality illusion to address non-stationarity

These changes address the 2 limitations of HAC, but they break HAC's technique to make training effective: addressing **the non-stationarity problem**, i.e. the problem that since all policies in the

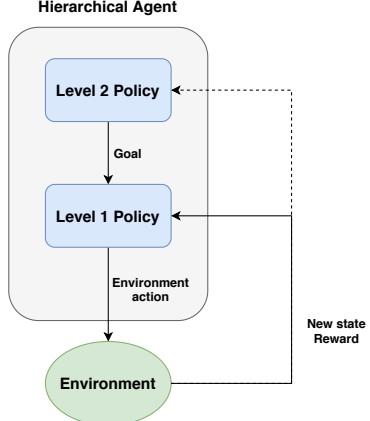

Figure 1: Interaction between a HAC-General hierarchical agent with 2 levels of policies and the environment. The goal-picking policy at the top produces goals and the goal-reaching policy at the bottom interacts with the environment to reach those goals in at most $H$ steps. The environment produces a reward and the state changes at each interaction. The objective of the goal-picking policy at the top is to pick goals that maximize the reward that is collected, while the objective of the goal-reaching policy at the bottom is to reach the goal states and collects at least the desired amount of reward dictated in the goal.

hierarchy train in parallel, each policy needs to continuously adapt to the changes in the policies below it in the hierarchy (whose behavior it relies on), which makes training difficult and unstable.

The insight of HAC is that if each policy trained under the illusion that all the policies below it were stationary, then it would train faster and more efficiently. Since optimal policies are stationary, **HAC attempts to give each policy the illusion that the policy below it is optimal** and thus stationary. HAC carefully constructs 3 types of transitions to create this illusion, where a transition is a tuple of the form *(state, action, reward, next state, goal, discount)*. While the 3 types of transitions are detailed in Appendix A for space reasons, we summarize how HAC creates the illusion that the policy below is optimal and how HAC-General With Teacher preserves that illusion.

We define some terminology: let $\pi$ be the policy in the hierarchy for which we create the illusion. We call $\pi$ the *goal-picking policy* since it produces goals and call the policy below it in the hierarchy the *goal-reaching policy* $\pi_{below}$, since it attempts to reach the goals it receives from $\pi$.

**HAC**. In HAC it is simple to give to $\pi$ the illusion that $\pi_{below}$ is optimal because rewards are only given when the goal-state is reached. As shown in Figure 2, if the action of $\pi$ is to pick the goal $g$ and the goal-reaching policy $\pi_{below}$ fails to reach it and reaches state $s$ instead, $\pi$'s action is replaced by the hindsight action $s$. The policy $\pi_{below}$ now appears optimal since $\pi$ picked goal $s$ and $\pi_{below}$ reached it.

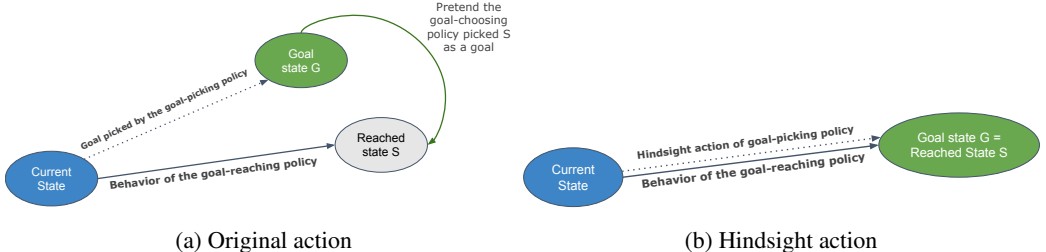

(a) Original action            (b) Hindsight action

Figure 2: HAC's optimality illusion: the goal-reaching policy $\pi_{below}$ doesn't reach the original action/goal $g$ but after replacing the original goal/action by the hindsight action $s$ it appears optimal.

**Problem**. This technique breaks down when environment rewards matter and must be maximized, *i.e.* it breaks down for HAC-General with Teacher. Replacing $g$ by $s$ is not enough anymore to give

the illusion that the goal-reaching $\pi_{below}$ acted optimally: while $\pi_{below}$ reached state $s$, it might not have collected the maximum amount of reward possible. In other words, there might be an alternative path to the same final state where more reward would have been collected (Figure 3). Since in most environments, it is impractical or impossible to determine if the optimal path was taken (or what the optimal path is), we cannot guarantee that $\pi_{below}$ appears optimal.

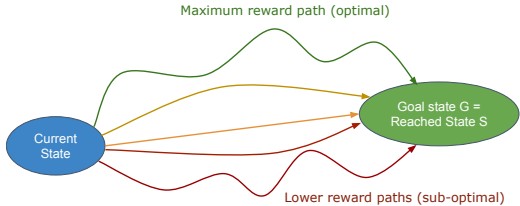

Figure 3: Now that rewards matter, reaching the goal isn't enough to guarantee optimality: the highest-reward path to the goal must be picked.

**Solution**. To address this issue, HAC-General uses a new definition of goals. The new goals have 2 components: a state $s$ which must be reached and a minimum amount of reward $r_{min}$ which must be collected. As shown in Figure 4, if the original action/goal is $(s, r_{min})$ but the goal-reaching policy reaches instead $s'$ and collects $r'$ reward, then the goal-picking policy's action is replaced by the hindsight action $(s', r_{hindsight})$ where $r_{hindsight} \leq r$, creating again the optimality illusion. It is important to note HAC-General creates the same 3 types of transitions as HAC; the major change is the way goals are defined.

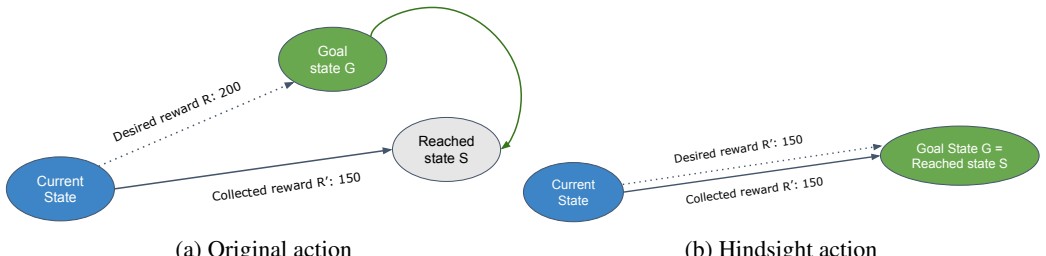

(a) Original action

(b) Hindsight action

Figure 4: HAC-General re-creates optimality by re-defining goals as a *(state, desired reward)* pair. If we replace the original action $(G, R)$ by the hindsight action $(S, R')$, the goal-reaching policy looks optimal again. Optimality is reached because optimality is relaxed: the policy must collect at least the desired reward to appear optimal, but not necessarily the maximum reward possible.

**Advantages**. The goal-picking policy now has 2 mechanisms to maximize the reward it collects[1]: (1) pick goal states $s$ that lead to high rewards and (2) force the goal-reaching policy to take a high-reward path to $s$ by making the minimum reward threshold $r_{min}$ as high as possible. The second point makes it possible to achieve high reward in RL environments where the reward is also tied to the action, not just the state, since the goal-reaching policy will learn to pick actions that lead to the goal-state **but also** lead to high reward.

## 3.2 Leveraging the Black-Box Agent

We can leverage the black-box agent to train the explainable agent more quickly and efficiently by letting the black box agent act as a teacher which provides partial demonstrations to the explainable hierarchical agent (the student). We stress that our method differs from classical imitation learning in two ways. First, the classical imitation learning algorithms are designed for non-hierarchical agents and cannot be directly applied for training hierarchical agents. Second, while a teacher is used to improve training, the objective that is being optimized in our work is not mimicking the expert, because that would create only a superficial link between the high level policy and the low

---

[1]Note: during the training process, the agent must learn to pick both good states and good desired rewards

level policy. Instead, all policies are still only optimized to maximize the reward (and we use the same reward scheme as before) but we occasionally create transitions from the expert's behavior.

Since the goal of this paper is to create an explainable agent, there is no requirement that it should be trained from scratch with no help, hence leveraging a black-box agent does not contradict our chosen setting.

During an episode, the goal-picking policy produces a series of goals and the goal-reaching policy tries to reach each of those goals. To train the goal-producing policy of the hierarchical agent, we need to obtain goals from the black box expert but the expert doesn't produce any goals explicitly since it is not a hierarchical agent. However, it produces goals implicitly: if we let the expert act for roughly $H$ steps, the last states it reaches are good goals. Additionally, the expert reached those final states using a series of actions $a_1, ..., a_n$ that can be used to train the goal-reaching policy.

Therefore, during an episode, each time a new goal needs to be picked, we stochastically decide whether the hierarchical agent or the black-box expert acts. If the black-box expert is chosen, it acts for a number $n \in [0.75H, H]$ of steps, reaching the state $s_n$ and collecting a total reward $r_n$. Given the last state we can create a goal $g_{hindsight} = (s_n, r_{hindsight})$ where $r_{hindsight} \leq r_n$. The algorithm then proceeds as usual, creating the transitions as if it was the goal-picking policy in the hierarchy that had picked the goal. To train the goal-reaching policy, for each action $a$ taken by the expert agent while it was reaching for $s_n$ we create a transition as if had been the goal-reaching policy acting with the short-term goal $g_{hindsight}$.

By interleaving the use of the hierarchical agent and the black-box expert during an episode, the behavior cloning problem is avoided (Ross & Bagnell, 2010) which ensures the hierarchical agent can achieve good performance even in states outside the expert-induced state distribution, *i.e.* even if it is in a state the expert would rarely or never visit when it interacts with the environment by itself (without the intervention of the hierarchical agent).

Algorithm 1 shows the pseudo-code to train a specific level (Appendix B contains the full pseudo-code). *Note*: the 3 types of transitions created in the algorithm are detailed in Appendix A.

---

**Algorithm 1:** Train-Level function of HAC-General (With Teacher)

---

1 **Input**: Initialized hierarchy of actor-critics with $k$ levels
2 **Input**: Teacher policy $\pi^*$ and probability C of using it
3 **Function** `TrainLevel` (*level* $l$, *state* $s$, *goal* $g, \pi^*, C$) **:**
4     $s_i, g_i, cumulR \leftarrow s, g, 0$
5     **repeat**
6        Decide if will test action/subgoal and then pick action $a_i$
7        **if** $l > 0$ **then**
8           **if** *useTeacherInIterationRandomly(C)* **then**
9              $s_i', r, \text{lowLevelTransitions} \leftarrow$ **expertRollout**$(s_i, \pi^*)$
10              $a_i \leftarrow [s_i', \textbf{reduceReward}(r)]$
11              Replay Buffer$_0 \leftarrow$ lowLevelTransitions
12           **else**
13              $s_i', r \leftarrow$ `TrainLevel` $(l-1, s_i, a_i, \pi^*, C)$
14        **else**
15           Execute primitive action $a_0$, observe next state $s_0'$ and reward $r$
16
17        $cumulR \leftarrow cumulR + r$
18        Create subgoal testing transition if failed to reach the subgoal and testing subgoals
19        Create the hindsight action and the hindsight action transition
20        Create incomplete hindsight goals transitions and put them in the HER buffer
21        $s_i \leftarrow s_i'$
22     **until** *episode ended **OR** ($H$ steps done or until any goal from above $g_{n,l \leq n < k}$ is reached)*;
23
24     Complete hindsight goal transition in buffer using HER and move them to the replay buffer
25     Return the final state $s_i$ and total reward $cumulR$

---

## 4   EXPERIMENTS

The goal of our experiments is to evaluate the effectiveness of the HAC-General With Teacher algorithm to train goal-producing hierarchical agents. We do not compare against most classical reinforcement learning algorithms and against imitation learning algorithms, since the former cannot train hierarchical agents and the latter creates no guarantee that the goal-reaching policy actually attempts to reach the goal, therefore not adding the interpretability we seek.

We compare 3 different algorithms: HAC-General with Teacher, HAC-General without Teacher, and the original HAC algorithm (to which we manually provide the end goal). We then use the trained hierarchical agent to generate explanations, which are visualized, showing the agent's plan to collect as much reward as possible.

We evaluate the agent in 2 environments from the OpenAI Gym (Brockman et al., 2016) which present different challenges: Mountain Car Continuous and Lunar Lander Continuous. In Mountain Car, a car has to reach the top of a mountain to its right but does not have enough power to reach it directly; it has to move left and then leverage momentum to climb the right mountain. The challenge of Mountain Car is good exploration since it only gets a positive reward at the top of the mountain but random exploration will very rarely lead to it; for other actions, it is punished through small negative rewards. In the Lunar Lander environment, a lander has to stabilize and land on the ground softly. This environment has more complex dynamics than Mountain Car, higher dimensional states (which makes picking good goals harder), and more complex actions for controlling the engines. The main challenge is predicting good states and desired rewards for the short-term goals, and learning how to reach those high dimensional goal states.

Figures 5 compares the performance of HAC, HAC-General Without Teacher, and HAC-General With Teacher. HAC achieves good performance in the Mountain Car (likely by knowing the end goal in an environment where exploration is the challenge) but fails to solve the more complex Lunar Lander environment (despite knowing the end goal). HAC-General Without Teacher only solves the Mountain Car environments on 3 out of 5 runs but always fails on Lunar Lander. HAC-General With Teacher can solve both environments, beating the other hierarchical algorithms and showing how effective leveraging a black-box expert can be to train the explainable hierarchical agent.

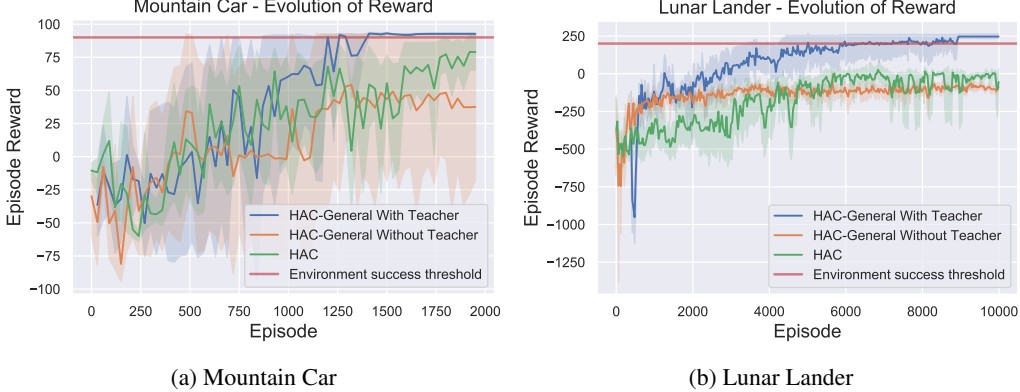

|(a) Mountain Car|(b) Lunar Lander|

Figure 5: Evolution of the performance of the agent during training in the Mountain Car and Lunar Lander environments. Each algorithm is executed 5 times per environment: the bold line represents the mean reward and the shaded regions display the standard deviation. HAC-General With Teacher performance exceeds the success threshold defined by the environment and outperforms both competing algorithms, showing both its effectiveness despite being in a more general and harder setting than HAC and the usefulness of leveraging a black box expert during training. *Note:* training stops once the success threshold is reached, since it indicates the agent can solve the task.

### 4.1 CREATING GOAL-BASED EXPLANATIONS

Given a trained hierarchical agent, we can leverage its goal-picking component to explain the agent's actions and future behavior. The current goal explains the short-behavior of the agent since it reveals the state the agent is trying to reach. We can obtain more useful explanations by querying for a series of goals instead of only knowing the current goal. These goals form the agent's plan to solve the environment and collect the maximum amount of reward; the goals help the user understand the agent's present and future behavior. To obtain this series of goals, the goal-picking policy is queried repeatedly: given the agent's state $s$, we query for the current goal $g_1 = (s_1, r_1)$; we then assume the agent reaches the state $s_1$ and query for the next goal $g_2 = (s_2, r_2)$; this process repeats until the desired amount of goals has been collected. Figure 6 displays the goal-based explanations at different time-steps of an episode, for both environments.

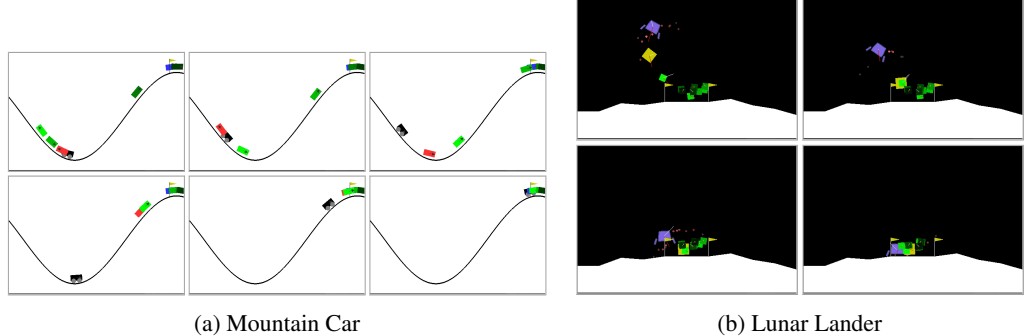

(a) Mountain Car                                  (b) Lunar Lander

Figure 6: Explanations produced by the hierarchical agent at different time steps of an episode. The red or yellow rectangle indicates the current goal. A plan composed of goals $g_1, g_2, ...$ is generated given the current state and shown as progressively darker green rectangles. A video version of the explanations is provided in the supplementary material.

In our experiments, the predicted goals are reached the majority of the time by the goal-reaching policy (see Appendix C), thus making the plans reliable indicators of the future behavior of the agent and useful interpretability tools.

## 5 PROPERTIES OF THE INTERPRETABILITY METHOD

**Dealing with failures of the goal-reaching policy.**    One advantageous property of the agent is that the plans can be created from any state $s$. Therefore, if at any point the goal-reaching policy fails to reach the goal by a large amount, interpretability will suffer since all the goals in the plan that come afterwards may not be valid anymore. However, performance may still be good because the goal-producing policy receives the current state when picking a goal and so can adapt to the errors made by the low-level policy.

**Why not simply run the simulator?**    While simply running the simulator also indicates the behavior of the agent, our method has advantages. It is for example advantageous in stochastic settings: we know at any time what the agent expects to be the best set of goals (before the stochastic event has happened) and understand how it adapts its the future goals as stochastic event happen and the agent ends up in states outside its initial goal plan.

**Orthogonality to other interpretability methods.**    The hierarchical agent add some interpretability due to the goals it produces, but a natural question arises: aren't the policies hard to interpret too? The strength of our method is that it is orthogonal to other interpretability techniques and can be combined with them and future progress in the field. Any method that can add interpretability to the neural networks underlying the policies could be applied to the policies of the hierarchical agent, making it possible not only to interpret better how the agent picks environment actions, but also to interpret better how high-level goals are picked, which might bring additional insight then in a non-

hierarchical agent. However, our method already brings some interpretability even without applying those other interpretability techniques; they complement each other instead of being in opposition.

**Goal granularity.** The number of intermediary goals is controlled through the maximum number of actions $H$ the goal-reaching policy may do to reach the goal before a new one is picked by the higher level policy picks a new one. A low value for H means goals are fine-grained and should be rapidly reached, while high values for H may lead to goals which require many more actions to reach and are therefore less granular.

## 6 CONCLUSION

In this paper, we tackle the problem of understanding the decisions taken by deep RL agents, which tend to be inscrutable black-boxes. We develop a new technique to understand agents, **goal-based interpretability**, in which the agent explicitly produces a series of goals and attempts to reach those goals one by one, collecting as much reward as possible along the way. These goals make the agent more explainable since the agent's current behavior is clear (reach the next goal) and the user knows the agent's long-term plan for solving the task. The agent becomes more interpretable since we know at any time the goals it seeks to reach and how this plan adapts as stochastic events happen in the environment. Additionally, obtaining the plan does not require taking any actions, contrarily to simulations, which is useful in real-world applications.

To create the goals, we rely on a 2-layer **hierarchical agent**, where the top layer produces goals and the bottom layer attempts to reach these goals. We contribute a new algorithm designed to train hierarchical goal-producing agents, which we call **HAC-General With Teacher**, which generalizes the Hindsight Actor-Critic (HAC) algorithm. Contrarily to HAC, HAC-General attempts to maximize the reward collected from the environment and doesn't require the environment to provide an end goal the agent must reach. HAC-General is also able to leverage black-box agents to improve the training process of the hierarchical agent. We detail possible improvements in Appendix D and note that the algorithm can be adapted to accommodate different definitions of goals, allowing for further research.

Our experiments show that HAC-General improves on HAC and can train goal-producing hierarchical agents faster and where HAC may fail to train. The trained hierarchical agent produces reliable and **clear explanations** which can be visualized, helping users understand the agent's short-term and long-term behavior. Additionally, our technique can be combined with other interpretabilty methods, adding more insight into how goals and environment actions are chosen.

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

## A    TRANSITIONS CREATED IN THE HINDSIGHT-ACTOR-CRITIC ALGORITHM

Figure 7 summarizes the transitions and we detail them below:

1. **Hindsight action transitions**: consider the scenario where the policy $\pi_l$ at level $l$ picked as action the goal $a_l = s$ and the policy below $\pi_{l-1}$ reached state $s'$. If $s'$ is close enough to $s$, then the goal was reached and the policy below $\pi_{l-1}$ acted optimally. If the goal wasn't reached, the action taken by the policy $\pi_l$ at level $l$ is substituted by the hindsight action $a_{hindsight} = s'$. Under this scenario, the policy below now acted optimally, since it reached the updated goal.

2. **Hindsight goal transitions**: for a policy $\pi_l$ at level $l$, while hindsight action transitions give the illusion that the policy below $\pi_{l-1}$ is optimal, the reward signal may still be extremely sparse if the policy $\pi_l$ never reaches the goal $g_l$ it received from above as input. To fix this issue, new transitions are created by replacing the goal $g_l$ of the hindsight action transitions by one of the states reached by policy $\pi_l$ while it was doing actions. This ensures that during at least one step of the episode the policy gets a positive reward since the hindsight goal is one of the states it visited.

3. **Subgoal testing transitions**: in the two previous types of transitions, the hindsight action is always a state that the policy below reached. Therefore, there are no transitions in which the action is a goal the policy below didn't reach in at most $H$ steps, which makes it impossible for the policy $\pi_l$ to learn which states are unreachable in $H$ steps. These unreachable states should not be picked, because then the policy below $\pi_{l-1}$ can never reach the goal it received, which will considerably slow down its training process and may lead to erratic and unstable behavior. To avoid picking unreachable goals as actions, we want to assign very low Q-values to those actions. However, there is one key uncertainty: if the policy $\pi_l$ produced the goal $g_l$ but the policy below $\pi_{l-1}$ reached the state $s$ instead, it might be either because the policy $\pi_{l-1}$ hasn't learned to reach the goal or because $g_l$ is unreachable in at most $H$ steps. Since it is not possible to distinguish between the cases with certainty, subgoal testing transitions are created a random fraction of the time (say 20%). In these transitions, the raw action is used (instead of the hindsight action) and a very low reward is attributed to that action, to avoid picking that goal again in the future.

*Note*: for the bottom-level policy, we only create hindsight goal transitions (using the original action). Since the bottom-level policy interacts with the environment instead of producing goals as actions, it is not possible to create hindsight action transitions and subgoal testing transitions.

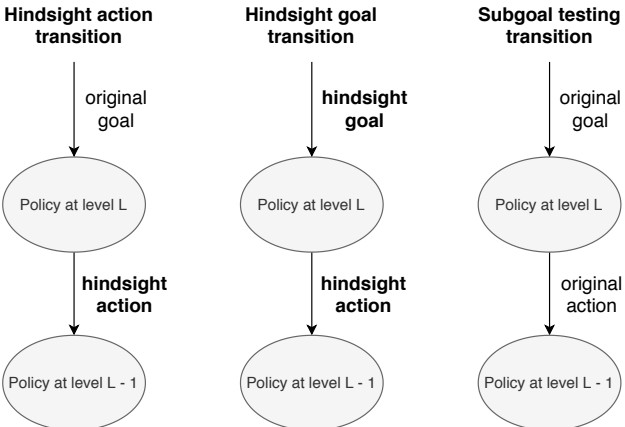

Figure 7: The three types of transitions created by the Hindsight Actor-Critic algorithm: hindsight action transitions (replace the action by the state that was reached by the policy below), hindsight goal transition (replace action; replace the goal by a state reached by the policy during the episode) and subgoal testing transition (if the policy at level L picks an action/goal the policy below fails to reach, occasionally attribute that action a very low reward).
.

## B  PSEUDOCODE FOR THE HAC-GENERAL WITH TEACHER ALGORITHM

The pseudo-code of the training loop and a more detailed version of the *TrainLevel* function is shown below. The HAC-General without Teacher algorithm can be obtained by setting $C = 0$.

The full implementation can be consulted in the supplementary material. The supplementary material also contains the implementation of the teacher agents as well as the code to show visualize the goals in the environment. For both environments, a GIF file shows the trained hierarchical agent acting, and both the current goal and future goals are visualized at all steps.

---

**Algorithm 2:** Train-Level (2-level hierarchy)

---

1  **Input**: Level $i$
2  **Input**: Start state $s$ and goal $g$
3  **Input**: Expert policy $\pi^*$
4  **Input**: Probability C of using the teacher
5  **Output**: Final state $s_i'$, total reward $cumulR$
6
7  $s_i, g_i, cumulR \leftarrow s, g, 0$
8
9  **repeat**
10      Decide if will test action/subgoal
11      $a_i \leftarrow \pi_i(s_i, g_i)$ (no noise $noise$ if subgoal testing)
12
13      **if** $i > 0$ **then**
14          **if** *useTeacherInIteration(C)* **then**
15              $s_i', r, \text{lowLevelTransitions} \leftarrow$ **expertRollout**$(s_i, \pi^*)$
16              $a_i \leftarrow [s_i', \textbf{reduceReward}(r)]$
17              Replay Buffer$_0 \leftarrow \text{lowLevelTransitions}$
18          **else**
19              $s_i', r \leftarrow \texttt{TrainLevel}(i-1, s_i, a_i, \pi^*, C)$
20      **else**
21          Execute primitive action $a_0$
22          Observe next state $s_0'$ and reward $r$
23      $cumulR \leftarrow cumulR + r$
24
25      **if** $i > 0$ *and subgoal $a_i$ was tested and wasn't reached* **then**
26          Replay Buffer$_i \leftarrow [s_i, a_i, Penalty, s_i', g_i, \gamma = 0]$ // Subgoal testing transition
27
28      **if** $i > 0$ *and $a_i$ wasn't reached* **then**
29          $a_i \leftarrow [s_i', \textbf{reduceReward}(r)]$ // Hindsight action
30
31      $R_t \leftarrow$ **getTransitionReward**$(s_i', cumulR, a_i)$
32      Replay Buffer$_i \leftarrow [s_i, a_i, R_t, s_i', g_i, \gamma \in \{\gamma, 0\}]$ // Hindsight action transition
33
34      // Hindsight goal transition (step 1)
35      **if** *not **isTopLevel**$(i)$* **then**
36          HER$_i \leftarrow [s_i, a_i, r = TBD, s_i', g = TBD, \gamma = TBD]$
37
38      $s_i \leftarrow s_i'$
39  **until** *episode ended **OR** (H steps done or until any goal from above $g_{n,i \leq n < k}$ is reached)*;
40
41  // Hindsight goal transition (step 2): perform HER; complete $r$, $g$ and $\gamma$
42  Replay Buffer$_i \leftarrow$ Perform HER(HER$_i$)

---

---

**Algorithm 3:** HAC-General (With Teacher)

---

**1 Input**: Initialized hierarchy of actor-critics with $k$ levels
**2 Input**: Expert policy $\pi^*$
**3 Input**: Probability C of using the teacher
**4**
**5 for** $M$ *episodes* **do**
**6**     $s \leftarrow reset\_environment()$
**7**     `TrainLevel`$(k - 1, s, goal = None, \pi^*, C)$
**8**     Update all the actor-critic networks in the hierarchy
**9 end**

---

## C PERCENTAGE OF GOALS THAT ARE REACHED

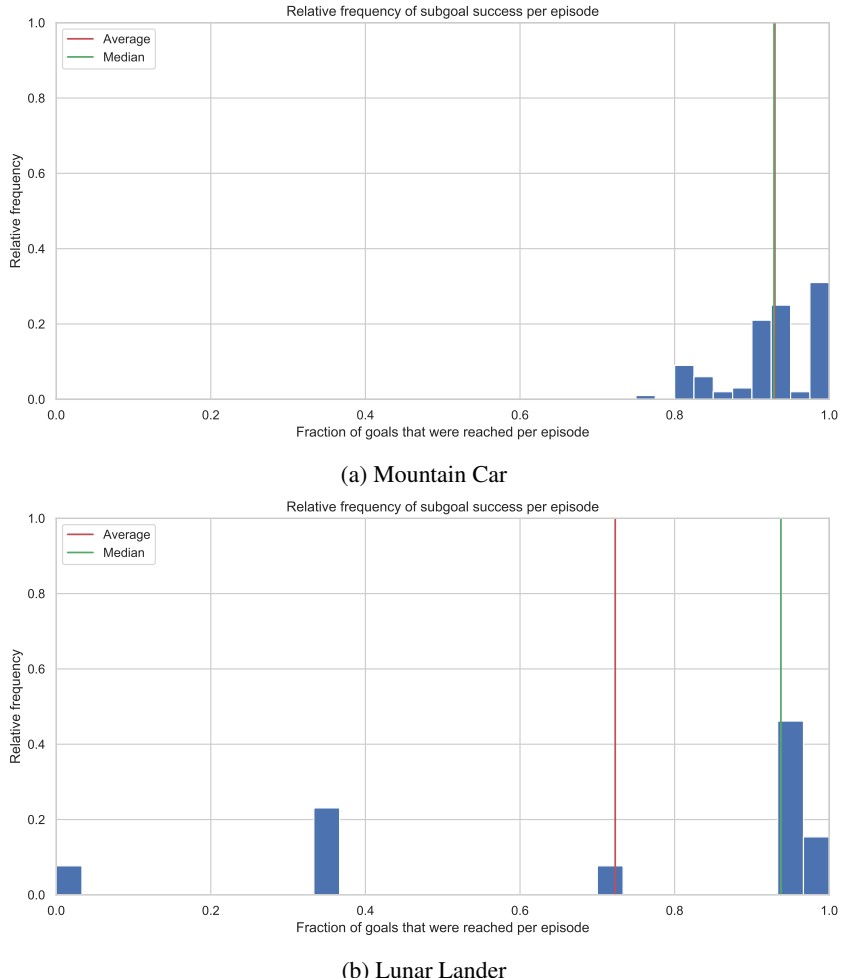

(a) Mountain Car

(b) Lunar Lander

Figure 8: The trained hierarchical agent runs for a hundred episodes and for each episode we measure the percentage of goals that were reached during the episode. The distribution of those percentages makes it possible to better understand how often the goals are reached. In the Mountain Car environment the goal is almost always reached (92 % median, 92% average) and in the Lunar Lander environment the goal is reached the majority of the time (96 % median, 74% average) despite goals being much more difficult to pick and reach.

## D    POSSIBLE IMPROVEMENTS

While we displayed explanations generated by the good-performance hierarchical agents and saw their effectiveness at gaining insight into the agent's decision-making mechanism and future behavior, improvements could be made:

1. While we generated plans at different time steps to validate them, this does not prove the plans are always effective (this is similar to the principle that tests can show the presence of bugs but cannot prove an absence of bugs). Finding mechanisms to further verify the reliability of the explanations would be interesting future work.

2. Work could be done to verify that the assumptions behind the plan-generating routine are met. Experiments show that the goal is reached the vast majority of the time for both the Lunar Lander and the Mountain Car environment (Figure 8), but more detailed examination could be useful. Additionally, we could also verify that the goal-picking policy isn't sensitive to small changes to the state it receives as input.

3. Training hierarchical agents in the setting where no goal is provided and the agent attempts to maximize the collected reward is a challenging task. Developing better algorithms to train hierarchical agents would make it possible to tackle more difficult environments and improve the efficiency and speed of the training process.

4. The experiments done in this paper did not deal with very high-dimensional states such as images. Predicting good states is likely very challenging in such environments and determining whether a goal has been reached or not becomes a more difficult question, so new techniques need to be developed for those scenarios, such as re-defining goals so that only the relevant part of the state is included in the goal (and background visuals are ignored, for example).

## E    HEURISTIC TO DETECT THE END OF A PLAN

One heuristic to determine how many goal-states should be queried when creating the plan is to detect when the goal-states stabilize. For example, when the lunar lander has reached the ground, the next goal should be to stay in the same state. Similarly, if the mountain car reaches the top of the mountain, the next goal should be to stay at the top of the mountain. Therefore, if we detect that the state-goal $s'$ produced by the goal-picking policy is very similar to the current state $s$, we may stop querying for new goals.

