# OpenReview forum: "Explainable Reinforcement Learning Through Goal-Based Interpretability"
_ICLR.cc/2021/Conference — Reject_

### Official Review · AnonReviewer2 · 2020-10-25
**Many details left out**

**Rating:** 3
**Confidence:** 4

**Review:**

The paper proposes a hybrid imitation learning/reinforcement learning method for learning hierarchical policies where the top layer provides sub-goals and desired cumulative rewards and the bottom layer learns to meet these goals. The advantage of such a decomposition is interpretability of the learned policy. The algorithm is evaluated on MountainCar and LunarLander from OpenAI’s gym. The authors show that their imitation learning/RL scheme is able to solve both tasks while producing reasonable sub-goals.

I found the sub-goal for interpretability idea very interesting. However the paper lacks in clarity when presenting the algorithm and in depth when evaluating it. For the presentation, the algorithm box should be commented and its main components should be explained in the main paper. Since most of the learning is carried away by HER (which is not even properly referenced in the algorithm box), a background section should be added. The paper should aim to be more self-contained. The idea of adding the cumulative rewards as a goal is easy to grasp and its explanation can be shortened, leaving space for a more thorough explanation of the actual learning of the hierarchical policy.

For the experiments part, important details are left out. For instance what is the ratio of imitation learning and reinforcement learning? How much imitation learning is needed for learning the task? How is the black-box policy obtained? What is its performance? To which extent the low level policy actually depends on the goals given by the higher level policy? Since imitation learning is involved for both layers, one could imagine that the top layer is learning to produce reasonable goals, the lower layer is learning to produce reasonable policies but there is only a weak connection between the two, and the lower level is only giving the illusion of tracking the top level’s goals.

I’m also unsure about some of the motivations. It is stated in the introduction that the drawback of black-box policies is that they can have surprising behaviors when facing unexpected states. How is the proposed policy immune to this phenomenon?

Finally, the number of intermediary goals should be discussed somewhere. In robotics, it is common to define the policy as generating a trajectory in joint space and to let a lower level position controller track the trajectory. One can also easily modify OpenAI’s Mujoco tasks to be position controlled and not torque controlled. Would a policy defined in this MDP, where an action is a desired joint configuration of the next state, be de facto interpretable, or is there some constraint in keeping the number of subgoals small to be human readable?

Overall, I like the direction and I think the plots showing the produced subgoals are very encouraging. However many details of the algorithm are left out and the experiments are not thorough enough to be fully convincing.

---

> ### Author Response · Authors · 2020-11-24
> **Response to reviewer 2**
>
> We thank the reviewer for their detailed comments and questions.
>
> > For the experiments part, important details are left out. For instance what is the ratio of imitation learning and reinforcement learning? How much imitation learning is needed for learning the task? How is the black-box policy obtained? What is its performance? To which extent the low level policy actually depends on the goals given by the higher level policy? Since imitation learning is involved for both layers, one could imagine that the top layer is learning to produce reasonable goals, the lower layer is learning to produce reasonable policies but there is only a weak connection between the two, and the lower level is only giving the illusion of tracking the top level’s goals.
>
> Our method is linked to imitation learning since there is a teacher that provides actions which are useful for learning, however we stress that it is distinct from imitation learning since no policy in the hierarchy is optimized to mimic the expert; they are optimized to maximize the total discounted reward. The low level policy is rewarded only for reaching goals, therefore its training process fully depends on the goals provided to it, since if learning is effective the goals produced by the high level policy will coincide with the implicit goals that could be deduced from the expert’s behavior by observing its actions.  In pure imitation learning, the concern that the lower level is only giving the illusion of tracking the top level’s goals is correct, as we discovered in non-described work where we attempted to extend Dagger to hierarchical agents. To ensure that there is a true link, we train the low level to reach the goals instead of simply picking similar actions (a much stricter and sparser reward setting) and additionally training only stops once the lower level policy consistently reaches at least a large fraction of the goals it receives.
>
> > I’m also unsure about some of the motivations. It is stated in the introduction that the drawback of black-box policies is that they can have surprising behaviors when facing unexpected states. How is the proposed policy immune to this phenomenon?
>
> We agree that our method is not immune to this phenomenon; however its strength is that this technique is orthogonal to other interpretability techniques and can be combined with them and future progress in the field. Any method that can add interpretability to the neural networks underlying the policies could be applied to the policies of the hierarchical agent, making it possible not only to interpret better how the agent picks environment actions, but also to interpret better how high-level goals are picked, which might bring additional insight then in a non-hierarchical agent. However, our method already brings some interpretability even without applying those other interpretability techniques; they complement each other instead of being in opposition.
>
> > Finally, the number of intermediary goals should be discussed somewhere. In robotics, it is common to define the policy as generating a trajectory in joint space and to let a lower level position controller track the trajectory. One can also easily modify OpenAI’s Mujoco tasks to be position controlled and not torque controlled. Would a policy defined in this MDP, where an action is a desired joint configuration of the next state, be de facto interpretable, or is there some constraint in keeping the number of subgoals small to be human readable?
>
> The number of intermediary goals is controlled through H, the maximum number of actions the goal-reaching policy may do to reach the goal before a new one is picked by the higher level policy picks a new one. A low value for H means goals are fine-grained and should be rapidly reached, while high values for H may lead to goals which require many more actions to reach and are therefore less granular.
>
> Regarding the Mujoco task, the interpretability brought by our technique depends on the interpretability of the states. If the mujoco joint configuration is easily understandable, either by direct examination, visualisation or other methods, then our method could bring insights to the user. Given the problem statement, the appropriate granularity H should be selected, to achieve the right balance between producing detailed plans and succinctness. This balance is subjective and so must be determined through human studies.

---

### Official Review · AnonReviewer1 · 2020-10-28
**It is hard to understand the method and how it compares against existing literature**

**Rating:** 3
**Confidence:** 5

**Review:**

The manuscript proposes (1) HAC-General with Teacher (GT), an extension of Hindsight Actor-Critic (HAC) that incorporates the environment reward as part of the goal formulation, and (2) "goal-based explanations", a framework in which the agent is tasked to produce intermediate goal states.

Good things:

+ The problem setting is important and interesting, and the paper lists a good number of applications where adding interpretability to policies would improve their applications to impactful real-life settings.
+ The idea of extending HAC to goal-less environments is also interesting, as it could potentially shine a method for systematically utilise HRL algorithms on standard RL environments.

Concerns:

1. The paper overall doesn't flow well, which makes it difficult to evaluate the novelty and quality of the method.

2. The concept of "goal-based explanations" is confusing. Most modern HRL work assumes the same problem decomposition, that that the agent needs to learn a goal-proposing policy (which might give additional signals such as rewards, embedding of the state, etc) and a acting policy that proposes environment-compatible actions. Thus it is unclear how this setup differs particularly from others.

3. The paper doesn't do a good job at explaining how looking at a series of goals (which in this case are just states) may be interpreted, and how these would correspond to "explanations". Note that I do not wish to be nitpickinging about this particular naming choice, but "explanations" almost raises the expectation that the end user would see some sort of dense (and possibly language-based) description of the policy.

4. GT builds on HAC, but HAC is never properly introduced. Section 3 can be mostly summarised as informing the reader that (a) HAC requires a hand-tuned goal proposal function, (b) that it doesn't utilise the environment reward (but it is not clarified why this would be a problem), and (c) that it has something to do with the problem of "non-stationarity". So, while section 3.1 does attempt to define then the differences between HAC and GT, it is extremely difficult to understand GT without clarifying how the manuscript intends HAC to look like.

5. The terminology used in Section 3 is extremely confusing. What is a "goal/action" (or "action/goal")? What does it mean for a policy to reach it? What is the difference between a normal action and a "hindsight action"? It would be good if all of these things were properly defined, and not left only to graphical form in figures 2, 3, and 4 (which all look extremely similar, and not that clarifying).

6. Section 3.2 declares that the goal of the work is to "create an explainable agent", and thus it is fair to use a pretrained teacher policy. However, (a) HAC does not require such a training setup, so it's unclear how to fairly assess whether GT is an improvement over it, and (b) it brings the method much closer to imitation learning, which the paper does not review at all. There are in fact methods that distill this sort of knowledge from teachers (e.g. https://arxiv.org/abs/1803.03835, https://arxiv.org/abs/1511.06295 https://arxiv.org/abs/2002.08037), and the proposed method looks extremely similar to them. It is also unclear how such a teacher policy would be trained to begin with, and how GT deals with e.g. probable sub-optimality of this teacher.

7. The experimental setting can be greatly improved. The paper proposes to test GT against HAC on mountain car and lunar lander, but (a) both environments are extremely easy to solve, and (b the difference in performance is probably due to HAC being presented an extremely sparse reward setting.

Due to these concerns, I currently cannot recommend acceptance, however I'd be willing to chance my score if the authors were to improve the manuscript by:

- Providing a detailed explanation of HAC, and how GT fundamentally _improves_ it towards achieving better interpretability.
- Providing a comparison (at least narrative-wise) against modern imitation learning literature.
- Improve the experimental setup by:
  a. At least having another modern RL algorithm as a baseline;
  b. Providing a fairer adjustment to HAC (or at least making an argument on how it is currently fair);
  c. Attempting a less trivial environment.

---

> ### Author Response · Authors · 2020-11-24
> **Response to reviewer 1**
>
> We thank the reviewer for the detailed feedback, as well as the references and desired improvements.
>
> > Problem decomposition
>
> We rely on the same problem decomposition, with the particularity that the environment does not provide a goal state that should be reached, which is required by some modern HRL algorithms such as the HAC algorithm we build upon.
>
> > The paper doesn't do a good job at explaining how looking at a series of goals (which in this case are just states) may be interpreted, and how these would correspond to "explanations". Note that I do not wish to be nitpickinging about this particular naming choice, but "explanations" almost raises the expectation that the end user would see some sort of dense (and possibly language-based) description of the policy.
>
> Explainability and interpretability are often used interchangeably and we were unaware that “explanation” carried this underlying expectation, so in this setting it is true that increased interpretability might be a more apt description of the objective of our work, since the key objective of our work is not to attempt to summarize the whole policy in a dense format. We thank the reviewer for their feedback.
>
> > What is a "goal/action" (or "action/goal")? What does it mean for a policy to reach it? What is the difference between a normal action and a "hindsight action"?
>
> The high level policy acts by choosing goals for the goal-reaching policy; therefore its actions correspond to goals. The hindsight action is the action that would have made the goal-reaching policy’s actions optimal, which can only be defined in hindsight once the goal-reaching policy has acted repeatedly to reach the goal. We thank the reviewer for this feedback on the writing.
> Section 3.2 declares that the goal of the work is to "create an explainable agent", and thus it is fair to use a pretrained teacher policy. However, (a) HAC does not require such a training setup, so it's unclear how to fairly assess whether GT is an improvement over it, and (b) it brings the method much closer to imitation learning, which the paper does not review at all. There are in fact methods that distill this sort of knowledge from teachers (e.g. https://arxiv.org/abs/1803.03835, https://arxiv.org/abs/1511.06295 https://arxiv.org/abs/2002.08037), and the proposed method looks extremely similar to them. It is also unclear how such a teacher policy would be trained to begin with, and how GT deals with e.g. probable sub-optimality of this teacher.
>
> (a) GT is an improvement over HAC for 2 reasons: (1) it can be applied to environments which do not provide goal states, such as the ones used in our evaluations. It was only possible to run HAC in those environments because we manually deduced the goal state and provided it to the HAC agent, and (2) in our experiments HAC was unable to solve the Lunar Lander environment while GT could, showing that leveraging an expert teacher is useful for training in environment with some constraints such as reward sparsity.
>
> (b) We differ from policy distillation methods and classical imitation learning in two ways. First, the standard description of those methods are designed for non-hierarchical agents and cannot be directly applied for training hierarchical agents. Second, while a teacher is used to improve training, this method differs from the traditional imitation learning setting since the objective that is being optimized in our work is not mimicking the expert, because that would create only a superficial link between the high level policy and the low level policy. Preliminary experiments showed that extending Dagger to hierarchical agents lead to good performance but no additional interpretability since the low level policy often failed to reach the goals set by the high level controller. Our hypothesis for this behavior is that the low level is never explicitly trained to reach the goal and the proxy objective of mimicking the expert is not sufficiently strict for goal-reaching to be learned consistently.
>
> > Further experiments
>
> We agree with the reviewer that further experiments would be useful to properly assess the relative effectiveness of GT against other goal-producing hierarchical algorithms. As the reviewer states, Lunar Lander is an easy environment to solve yet HAC is unable to solve it in our experiments (and HIRO [1] as well in our preliminary experiments which were not described). As a first step required before contemplating more difficult environments, our experiments were done to check if an expert teacher would help alleviate this problem or not, and the results show it does help hierarchical learning in these settings.
>
> We thank the reviewer for taking the time to provide precise and useful improvement steps.
>
> [1] Nachum, O., Gu, S. S., Lee, H., & Levine, S. (2018). Data-efficient hierarchical reinforcement learning. In Advances in Neural Information Processing Systems (pp. 3303-3313).

---

### Official Review · AnonReviewer4 · 2020-10-31
**Official Blind Review #4**

**Rating:** 4
**Confidence:** 4

**Review:**

This paper proposes a hierarchical RL method where the high level controller produces a series of sub-goals in an open-loop fashion which the low-level controller attempts to reach sequentially with the aim of maximising task rewards. The agent is trained using an extension of Hindsight Actor-Critic (HAC) algorithm. The algorithm also leverages a model-free flat policy trained on task rewards as an expert. The approach is evaluated on two tasks: Mountain Car and Lunar Lander.

The related work section lacks a thorough analysis of existing literature in this domain. Additionally, the lack of baselines is quite concerning, I would have liked to see a comparison with closed loop hierarchical agents, or model based hierarchical agents.

1. While the motivation of the work is interesting, the paper lacks novelty and fails to provide any comparisons with existing HRL methods that could also directly fit in the category of explainable RL methods. There have been several papers attempting sub-goal discovery in HRL which are not discussed in this work. Hence, it is hard to assess the novelty and how it stands in comparison to existing methods. Some recent work in this direction: ([Jurgenson et. al., 2019](https://arxiv.org/abs/1906.05329), [Parascandolo et al., 2019](https://arxiv.org/abs/2004.11410), [Nasiriany et al., 2019](https://arxiv.org/abs/1911.08453))

2. To my understanding, the method only works if the low-level controller has near optimal performance as otherwise setting open-loop goals would not be plausible. For example, the agent might diverge significantly from the path expected by the high level-controller while reaching one of the subgoals and hence the rest of the plan is not relevant anymore. This limits the usefulness of this method significantly, making it only plausible in deterministic settings and it is unlikely that it would generalise outside of the distribution of data seen during training. Could you comment on how this could be addressed?

3. Connected to that, other work on model-based HRL also seems directly relevant in this context where there is no need to assume necessarily that the low-level controller is perfect as the high level transition model is learned and leveraged during planning to find a most rewarding series of subgoals. This has an additional benefit of not assuming the low-level policy is perfect and rely on a learned transition model of behaviour for the agent. It also does not rely on having access to an expert low-level controller to provide demonstrations. It would be nice to better motivate the reasoning behind assuming a perfect low-level controller and discuss its failure cases. Relevant works in this direction include  an ([Nasiriany et al., 2019](https://arxiv.org/abs/1911.08453 ), [Pierrot et al, 2020](https://arxiv.org/abs/2007.13363))

4. It is not clear to me how many goals are queried from the high level controller at the start of an episode, the paper states “...,this process repeats until the desired amount of goals has been collected.” It would be good to discuss this in the main text.

In summary, while the motivation seems sound and important, the problem the paper is addressing is not particularly novel. At the same time, the assumptions made seem quite restrictive to me, the experiments are not particularly convincing and rigorous comparisons to baselines are missing.

Hence unfortunately, I don’t believe this paper, in its current form, is appropriate for publication. But I encourage the authors to improve their related work section, discuss and compare their method with existing approaches to subgoals discovery and model-based HRL to better motivate the novelty and significance of this work.

---

> ### Author Response · Authors · 2020-11-24
> **Response to reviewer 4**
>
> We thank the reviewer for their detailed feedback, questions and references.
>
> > To my understanding, the method only works if the low-level controller has near optimal performance as otherwise setting open-loop goals would not be plausible. For example, the agent might diverge significantly from the path expected by the high level-controller while reaching one of the subgoals and hence the rest of the plan is not relevant anymore. This limits the usefulness of this method significantly, making it only plausible in deterministic settings and it is unlikely that it would generalise outside of the distribution of data seen during training. Could you comment on how this could be addressed?
>
> While we agree that in highly stochastic environments it is almost impossible to provide a plan that never diverges from the actual behavior (which will depend on the stochastic events occurring in the environment), we believe that the method remains useful to determine the states the agent determines to be good goal states that balance high short term reward, high long term reward and high probability of reaching that goal state, therefore bringing some interpretability into its long term plans in the face of large uncertainty.
>
> The challenge of good generalisation for hierarchical agents is a difficult one, which we have attempted to address by relying on an expert teacher. Our work has the dual objective of training a goal-producing hierarchical agent and leveraging those goals as interpretability tools. While non-optimal performance of the low-level controller may hinder the ability to perform the latter, it might not be an issue for the overall performance of the agent since the high level controller receives the state when choosing its action and thus can adapt to the failures of the policy below it. Since the high-level policy produces actions by sampling from a distribution, multiple paths could be obtained by sampling several other high-likelihood goals and computing the plan corresponding to each of those goals states (this may be done recursively to some extent), showing not only the path the high-level policy views as best but also paths are attributed non-negligible probabilities (thus showing how the agent would react to different stochastic or unexpected events).
>
> > Q: Assumption of a perfect low-level controller.
>
> While good performance of the low-level controller is required for the explanation to correspond to the actual behavior, but it is not necessarily required for the agent to have good performance, since the high-level policy receives the current state when picking a goal and so can adapt to the errors made by the low-level policy.
>
> > It is not clear to me how many goals are queried from the high level controller at the start of an episode, the paper states “...,this process repeats until the desired amount of goals has been collected.” It would be good to discuss this in the main text.
>
> Currently, a predefined number of goals is queries per environment, but as discussed in the appendix more sophisticated approaches could be taken, such as stopping querying when the goals states stabilise (possibly indicating that the state is terminal). Determining the optimal amount of goals is not the core problem of our work, and we leave detailed evaluations of different querying techniques for future work.

---

### Official Review · AnonReviewer5 · 2020-11-04
**Not enough evidence for the claims**

**Rating:** 3
**Confidence:** 4

**Review:**

Summary

This paper proposes a hierarchical reinforcement learning algorithm in an attempt to improve the explainability of RL agents via the goals proposed by the higher level policy.

Strengths

The problem of designing safer and more transparent RL agents is an important and rather neglected one, so it is nice to see papers on this topic.

Weaknesses

However, I think the paper makes some unsubstantiated claims, lacks thorough empirical evaluations, and is not very well-motivated and situated in the broader RL literature.

First of all, the proposed approach is only evaluated on two very simple tasks (Mountain Car and Lunar Lander) and it is only compared with two ablations of the proposed method. At the very minimum, the paper should include comparisons with a strong RL baseline (i.e. SAC, PPO etc.), a strong HRL baseline (i.e. feudal networks), and a behavioral cloning baseline (i.e. vanilla BC, Dagger, or GAIL) since the algorithm includes elements of all these. I would be quite surprised if SAC and BC don’t achieve similar or much better performance than the proposed methods on these simple environments. Given that you have access to an expert, it is really unclear to me why you would want to use your method versus simple behavioral cloning. There needs to be better  motivation.

I also do not think the choice of environments is well-motivated. Besides the fact that they are quite simple for current SOTA, they do not strike me as particularly good for emphasizing explainability or for using a hierarchical agent (since they can be easily solved without HRL). I also don’t understand how the goals are useful in this particular case. Given that these are fixed environments, once you’ve trained a RL policy you can accurately predict the agent’s trajectory including all the visited states rather than only some of them (as the proposed method does). I think a more interesting use case would be to test the method on a new environment and show the goals selected by the higher level policy since a typical RL method wouldn’t be able to predict what states it will visit in a new environment without a simulator or running the policy. I suggest using environments that are better suited for your goals and approach and where other methods might fail or would be hard to explain.

In section 4.1 you make claims regarding the “resistance to small errors” and to “dynamics” / stochasticity which are not supported by any experiments or theory. I suggest either removing them or backing them up with some empirical evidence.

Finally, I think the core idea of the paper needs to be better motivated. After reading the paper, I am not fully convinced that the proposed architecture to generate goal-based explanations is very useful for understanding and supervising an agent’s behavior. If you consider the environment in which you are training, then I don’t see how predicting goals is better than simply using a standard RL to predict actions and use the simulator to predict future states. If you consider a new environment, I don’t see how the proposed method can generalize if the state space is different so it won’t know what goals to output or its predictions will be off.

Also, what is the level of granularity for generating the goals (i.e how often the higher level policy acts relative to the lower level policy) and is there a way to tune this? This seems like an important design choice which is not discussed very much in the paper.

Recommendation

Given the above points, I do not think the paper is ready for publication.

---

> ### Author Response · Authors · 2020-11-24
> **Response to reviewer 5**
>
> We thank the reviewer for their detailed feedback.
>
> > First of all, the proposed approach is only evaluated on two very simple tasks (Mountain Car and Lunar Lander) and it is only compared with two ablations of the proposed method. At the very minimum, the paper should include comparisons with a strong RL baseline (i.e. SAC, PPO etc.), a strong HRL baseline (i.e. feudal networks), and a behavioral cloning baseline (i.e. vanilla BC, Dagger, or GAIL) since the algorithm includes elements of all these. I would be quite surprised if SAC and BC don’t achieve similar or much better performance than the proposed methods on these simple environments. Given that you have access to an expert, it is really unclear to me why you would want to use your method versus simple behavioral cloning. There needs to be better motivation.
>
> We agree with the reviewer that more extensive experiments should be done on more complex tasks in order to support the claims better, but due to time limitations this was not possible. However, we stress that we do not claim  the HAC with Teacher is more sample efficient or achieves the maximum performance when compared to other general algorithms; instead, our claim is that it allows training **hierarchical goal-producing** agents in some settings, with the goal-producing property being key to add some degree of interpretability to the agent.
>
> While the classical RL baselines (SAC, PPO) or behavior cloning techniques (BC, Dagger, GAIL) are more sample efficient or may lead to higher performance, in the standard formulation that are not directly usable to train hierarchical agents where some policies of the hierarchy produce goals, therefore not being applicable with our interpretability approach which relies on goal-producing agents. While goals produced by feudal networks [1] might be more effective for training, they do not fit our interpretability objectives either since the goal space is a latent space that is not directly understandable to researchers and non-experts.
>
> We have attempted to use behavior cloning techniques to train the hierarchical agents but rapidly discovered that while training was fast and the agent had good performance, the agent was not more interpretable because the goal-reaching policy often failed to reach the goal. While we would need to do extensive experiments to confirm this, our tentative explanation for this problem is that in behavior cloning the optimization objective for the goal-reaching policy is to mimic the expert, which is similar but distinct from the objective that matters for our application: reaching the goal.
>
> >In section 4.1 you make claims regarding the “resistance to small errors” and to “dynamics” / stochasticity which are not supported by any experiments or theory. I suggest either removing them or backing them up with some empirical evidence.
>
> We agree with the reviewer regarding the “resistance to small” and have removed it. The “dynamics” claim is not an empirical one but a theoretical one; written differently, it states that it is possible to compute goals in any state (even if the agent isn’t in that state) and without running the simulator for the environment, or without running the system in the real world.
>
> > Motivation for the paper
>
> We thank the reviewer for this important feedback and agree that in the specific environments of the experiments simulations could be an effective alternative, though it has limitations. The objective of the experiments was to demonstrate the possibility to train hierarchical agents using expert agents as aids for faster learning and then evaluating the quality of the resulting hierarchical agents, as an important step preliminary to extensive evaluations on more complex environments, which were left for future work due to time limitations. One example of the advantage of our method over simple simulations is in stochastic settings, as we interpret better what the agent expects to be the best goal path at any time and understand how it adapts its goal plan as stochastic event happen and the agent ends up in states outside its initial goal plan. We agree that directly applying the agent in a new environment with a new state space will not necessarily work directly and further learning may be required.
>
> > Also, what is the level of granularity for generating the goals (i.e how often the higher level policy acts relative to the lower level policy) and is there a way to tune this?
>
> The granularity is controllable through H, the maximum number of actions the goal-reaching policy may do to reach the goal before a new one is picked by the higher level policy picks a new one. A low value for H means goals are fine-grained and should be rapidly reached, while high values for H may lead to goals which require many more actions to reach and are therefore less granular.
>
> [1] Vezhnevets, Alexander Sasha, et al. "Feudal networks for hierarchical reinforcement learning." arXiv preprint arXiv:1703.01161 (2017).

---

### Author Response · Authors · 2020-11-24
**General Comments**

We thank the reviewers for their detailed comments. We have updated the paper where it was unclear or there were typos. However, we did not have time to address all the comments, such as the suggestion to perform more experiments, for this conference.
As suggested by the reviewers, we will now consider that our main contribution is a Reinforcement Learning agent that, in addition to learning a task, also learns subgoals to achieve the task. The main benefit of our approach is that it can be used to show the users a sequence of subgoals, or waypoints, that the agent will try to reach in the future. Contrary to other approaches, our subgoals are in the agent’s state-space, so possible to present to the user, instead of some learned feature space. Being able to predict future waypoints is also important in real-world applications, in which no simulator can be used to “roll-out” the agent and show the user what will happen in the future. It is also a useful technique to evaluate how the agent adapts the desired waypoints  when an unexpected stochastic event happens and the agents is in a state not predicted by its plan.

Our paper also considered the use of our subgoal learning architecture to improve the quality of the policy learned by the agent. We will decrease the emphasis on that, and instead focus on producing sequences of waypoints, to increase the trust an user has in an agent by being able to see what it will do in the future. The reviewers also suggest that we perform more experiments, in more challenging environments. We will do that in the near future, and submit a revised paper to another conference. We point out that by focusing on our main contribution, the possibility to produce waypoints, instead of also discussing the sample-efficiency of our agent while learning, we drastically reduce the number of environments and baseline algorithms to compare our approach to (mentioned by Reviewer 5). We will focus on environments in which waypoints are beneficial, and compare our algorithm to subgoal-identification baselines ([Jurgenson et. al., 2019](https://arxiv.org/abs/1906.05329), [Parascandolo et al., 2019](https://arxiv.org/abs/2004.11410) , [Nasiriany et al., 2019](https://arxiv.org/abs/1911.08453)).

---

### Decision · Program_Chairs · 2021-01-07
**Final Decision**

**Decision:**

Reject

**Comment:**

This paper proposes a method for hierarchical decision making where the intermediate representations between levels of the hierarchy are interpretable. I personally really like this general direction, as did most of the reviewers. Unfortunately, it was felt that, even after discussion, this paper is not ready for publication. To summarize the general spirit of the objection to this paper, all reviewers found that the experimental section was faulty and did not match the claims of the paper. Specifically, criticism here surrounded first, the choice of experimental setting, which was not considered to be the best for testing interpretable hierarchical decision making approaches; and second, the choice of comparison/baselines, which did not give sufficient security that the results produced by the proposed approach were sufficiently impressive.

I am satisfied that the reviewers considered the paper fairly, gave constructive criticism, and took onboard the author feedback. As a result, I am recommending rejection. I nonetheless think that the high quality feedback provided here will enable the authors to prepare follow-up experiments that may show their method in a more robust positive light, and encourage them to submit to a future conference, once armed with such results.